# Enteral Nutrition by Nasogastric Tube in Adult Patients under Palliative Care: A Systematic Review

**DOI:** 10.3390/nu13051562

**Published:** 2021-05-06

**Authors:** Eduardo Sánchez-Sánchez, María Araceli Ruano-Álvarez, Jara Díaz-Jiménez, Antonio Jesús Díaz, Francisco Javier Ordonez

**Affiliations:** 1Internal Medicine Department, Punta de Europa Hospital, Algeciras, 11207 Cádiz, Spain; 2Instituto de Investigación e Innovación Biomédica de Cádiz (INiBICA), Hospital Universitario Puerta del Mar, Universidad de Cádiz, 11009 Cádiz, Spain; 3Punta de Europa Hospital, Algeciras, 11207 Cádiz, Spain; malva757@gmail.com; 4Faculty of Education Sciences, University of Cádiz, 11519 Puerto Real, Spain; luna_nueva17@hotmail.com; 5Medicine Department, School of Nursing, University of Cadiz, Plaza Fragela s/n, 11003 Cadiz, Spain; antoniojesus.diaz@uca.es; 6Human Anatomy, School of Medicine, University of Cádiz, Plaza Fragela s/n, 11003 Cadiz, Spain; franciscojavier.ordonez@uca.es

**Keywords:** artificial nutrition, enteral nutrition, nasogastric feeding, nasogastric tube, palliative care

## Abstract

Nutritional management of patients under palliative care can lead to ethical issues, especially when Enteral Nutrition (EN) is prescribed by nasogastric tube (NGT). The aim of this review is to know the current status in the management of EN by NG tube in patients under palliative care, and its effect in their wellbeing and quality of life. The following databases were used: PubMed, Web of Science (WOS), Scopus, Scielo, Embase and Medline. After inclusion and exclusion criteria were applied, as well as different qualities screening, a total of three entries were used, published between 2015 and 2020. In total, 403 articles were identified initially, from which three were selected for this review. The use of NGT caused fewer diarrhea episodes and more restrictions than the group that did not use NG tubes. Furthermore, the use of tubes increased attendances to the emergency department, although there was no contrast between NGT and PEG devices. No statistical difference was found between use of tubes (NGT and PEG) or no use, with respect to the treatment of symptoms, level of comfort, and satisfaction at the end of life. Nevertheless, it improved hospital survival compared with other procedures, and differences were found in hospital stays in relation to the use of other probes or devices. Finally, there are not enough quality studies to provide evidence on improving the health status and quality of life of the use of EN through NGT in patients receiving palliative care. For this reason, decision making in this field must be carried out individually, weighing the benefits and damages that they can cause in the quality of life of the patients.

## 1. Introduction

Initially, the aim of Palliative Care (PC) was to relieve suffering at the end of life. However, it is nowadays considered as a model to follow in patients in whom there is no curative treatment, and is therefore being implemented at earlier stages. Initially, PC was focused on cancer patients, but it currently covers other conditions such as advanced dementia, HIV/AIDS, heart disease, etc. [1].

Every year, 40 million people need palliative care, but only 3 million have access to such special attention [2]. Currently, the goal of PC is to promote comfort and to maintain an optimal quality of life for patients and their families under palliative care [3] through prevention and management of physical, psychosocial and spiritual issues in these patients [4]. It should not be forgotten that quality of life evaluates the subjective perception that each patient has around alterations or limitations that the disease undertakes in the physical, psychosocial and spiritual aspects of their lives [5].

Nutrition and hydration are basic elements for maintaining life, and they are considered signs of health in our society [6]. Occasionally, patients present failure at maintaining adequate oral intake for meeting their nutritional needs [7], and this can lead to physical and psychosocial issues such as anxiety and distress [8]. Therefore, it may be necessary to commence Artificial Nutrition (AN). In 2008, Cochrane published a review regarding the use of AN in adult patients receiving palliative care. The authors concluded that there was not enough evidence to guide the development of guidelines for practice [9]. Six years later, the update of this review presented the same results; therefore, there are no new quality studies regarding this subject [7].

If the patient takes less than 50% of their nutritional requirements and there are no contraindications or bronchoaspiration risks, and their life expectancy is less than 6 weeks [10], Enteral Nutrition (EN) must be prescribed through a nasogastric (NG) tube [11]. This is a widely used and easily accessible technique, although in the case of patients with advanced dementia who receive PC, evidence supporting the use of NG tube is limited, and this technique may have a negative impact on the quality of life of these patients [12]. The use of tubes in patients with advanced dementia does not improve survival, prevent aspiration [13], or improve their functional status. In addition, the use of tubes for artificial nutrition has been associated with agitation, increased physical restrictions, and complications related to the tubes [14,15].

The use of EN through NG tubes in patients under PC continues to be a controversial subject [16], since there is little evidence on the role of nutritional support and whether its implementation improves quality of life. In addition, it affects the psychological sphere of patients, because it can influence their social relationships and the way they interact with others. However, Mitchell et al. reported that more than a third of nursing home residents with dementia had been subjected to a feeding tube [17]. Decisions and/or choices may confront patients, family members, and health professionals. Therefore, having a good knowledge of the benefits and harms of the use of this technique is paramount in order to reduce ethical conflicts and to understand how the use of this technique can influence the physical, psychological and spiritual spheres, and therefore, the quality of life of patients receiving PC. Accordingly, the goal of the present study is to understand the current state of the management of EN using NG tubes in patients receiving palliative care, along with its effect on health status and quality of life.

## 2. Materials and Methods

A systematic review of the literature was made. The results were obtained by direct online access through the following database: PubMed, Web of Science (WOS), Scopus, y Scielo, Embase y Medline. The aim of this review was to address the next question: Is it appropriate the use of EN by NG tube in patients under palliative care?

To define the research, question the PICOS criteria (Table 1) was used.

The articles reviewed were published in any country, by any institution or individual investigator, and written in Spanish or English. The research was limited to those published in the last 5 years (between 2015 and 2020).

For the documentary retrieval, the following MeSH descriptors were used: “palliative care”, “enteral nutrition”, “terminal care” “terminally ill”. Neither Subheadings nor Entry Term classifiers were used. The search strategy was: (“Palliative Care” OR “Terminal Care” OR “Terminally ill”) AND “Enteral Nutrition”. The final choice of articles was made following the inclusion criteria: (a) studies published in journals indexed in international databases subject to peer review, (b) published between 2015 and 2020, and (c) written in English or Spanish; and the exclusion criteria were: (a) studies based on pediatric age, (b) expert reports, editor’s letters, books, monographs, clinical narratives or reviews. Due to the large number of articles found in the first search, and as a quality assessment, two screenings were carried out. The first was based on the title and summary, eliminating those articles that dealt with a topic other than the one proposed. In the second screening, review articles, editor’s letters, etc., were eliminated.

To carry out the critical reading and evaluation of the articles found, the STROBE (Strengthening the Reporting of Observational studies in Epidemiology) statement was used for the observational studies [18] and the CONSORT guide (Consolidated Standards of Reporting Trials) for randomized clinical trials [19].

## 3. Results

A total of 403 articles were found: 32 (7.9%) PubMed, 30 (7.4%) WOS, 20 (4.9%) Scopus, 4 (1.0%) Scielo, 151 (37.4%) Embase, and 166 (41.2%) Medline. Of these recovered papers, 223 (55.3%) were redundant.

Once the first screening was applied based on the title and abstract, 168 articles were eliminated. After the second screening, nine articles were eliminated. The number of articles selected was three, all of which were observational studies, for which the STROBE statement was made. All of these articles fulfilled 90% of the points of the set declaration. The parameters of PRISMA (Preferred Reporting Items for Systematic Review and Meta-Analyses) were followed (Figure 1).

The results obtained showed different study parameters in the approach to the proposed topic (Table 2). No studies were found that addressed the use of NGT versus not using a feeding tube, but there was always a third group representing the use of either Percutaneous Endoscopic Gastrostomy (PEG) or esophageal stent. Therefore, the results obtained in relation to the use of NG tube and the other groups were taken.

In the study carried out by Bentur et al. 2015 [20], three groups were compared: subjects without feeding tubes, subjects with NG tubes and another group caring Percutaneous Endoscopic Gastrostomy (PEG). The results related to the use of NG tube versus the non-use of a catheter or the use of PEG were taken as a reference for this review. They concluded that the use of a feeding tube in people with advanced dementia in the community was associated with negative outcomes and increased caregiver burden. The use of an NG tube caused less diarrhea and more restrictions than the group that did not carry a catheter. The use of feeding tubes increased attendances to the emergency department, although they did not distinguish between NGT and PEG. No statistical difference was found between catheter use (NG tube and PEG) and non-use with respect to the treatment of symptoms at the end of life, comfort or satisfaction at the end of life.

Yang et al., in 2015, compared hospital stays and survival among patients with esophageal obstruction and a short life expectancy in subjects with EN by tube, with esophageal stent placement, and with nutritional support without oral intake. The results obtained showed that the patients with NGT and esophageal stent had a shorter hospital stay (19 and 12 days, respectively) and a longer median survival (*p* < 0.01) than the group with nutritional support. Concluding that enteral feeding by NG tube in palliative care was safe, inexpensive, and had a low complication rate [21].

The multicenter study carried out by Shinozaki et al. in 2017 in Japan, found that 74.6% of patients in the terminal phase required EN.

These authors suggest that the nutritional intake route may play a role in quality of life. No significant difference was found in quality of life between the different study groups. However, the mean hospitalization period was significantly shorter for gastrostomy-fed patients than for nasogastric tube-fed patients (21 vs. 64 days). Patients with PEG had a shorter period between study prescription and death than patients fed through an NG tube [22].

## 4. Discussion

The results obtained show the limited bibliography in the field of EN through NG tube in patients receiving palliative care. There are studies on the use of tube feeding in these patients, but without distinction between the NG tube and PEG, so it was not possible to obtain individual and differentiated results between both routes of administration.

The articles in this research can be found to represent a low level of evidence, since they are observational studies, and no randomized clinical trials (RCTs) were performed. These results coincide with those reported by other studies, such as the systematic reviews carried out by Good et al. in 2008 and later in 2014 [7,9].

Malnutrition leads to increased comorbidities and decreased performance status and quality of life [10]. Therefore, nutritional support should be integrated into palliative care, and its implications with respect to quality of life and life expectancy should be assessed [23]. Within such nutritional support is included the use of nutrition through a tube, although its use remains controversial, especially in the case of the NG tube. The emergence of research and guidelines on the management of patients under palliative care has managed to reduce the use of tube feeding by 50% [24].

Some studies report that the use of enteral tube feeding is effective for improving the quality of life of patients [25], since it may improve physical, psychosocial and spiritual aspects. Although the quality of life of patients with NGT was not studied in the study carried out by Bentur et al. in 2015, they did find that these patients presented more diarrhea and restrictions, which can affect the physical and even psychosocial sphere, which could influence the quality of life of these patients.

Even though there was no distinction between patients with NG tube and PEG, it was concluded that these patients attended the emergency department more times than those who did not carry any type of feeding tube, which also negatively influences their quality of life, since they present more comorbidities, making it necessary for them to go to a health center more frequently, and causing changes in their daily life, as reflected in the well-being subscale of CAD-EOLED [20]. Another aspect that can negatively influence quality of life is the increase in the number of hospital stays and the decrease in survival. The use of NGT may decrease hospital stays and improve survival in patients receiving palliative care, and thus improve the quality of life perceived by these patients [21]. However, Shinozaki et al. concluded that subjects presenting NG tube had longer hospital admissions than those using PEG. Even though the survival period was longer, no significant differences in quality of life were found among the various groups [22]. This may be due to the choice of the measurement interval, since it was performed in patients with a short life expectancy. It should be noted that the perception of quality of life is related to reality and expectations. In patients receiving palliative care, the expectations for improvement are sometimes low, especially when their life expectancy is short [26]. Perhaps for this reason, no differences were found in quality of life in these investigations. The scant evidence on this topic has led to different interpretations and approaches in these patients.

The Ethics Work Group of the Spanish Society in Parenteral and Enteral Nutrition (SENPE) recently (2019) published a confirmation that the placement of tubes for nutrition in patients with advanced dementia was a futile treatment that only contributed to prolonged suffering and concluded that health care professionals should not make wide use of EN by tube [27]. Schwartz et al. considered that EN by tube could improve quality of life, but that the benefits in the last days phase were limited and did not exceed the loads [28].

Furthermore, there may be discrepancies between health professionals and patients when prescribing nutritional support through an NG tube. For example, Amano at al. found that 78.6% of subjects in their study did not wish to receive artificial nutrition by feeding tube, even though their intake was insufficient [29]. In the study undertaken by Pengo et al. in 2017, it was found that the numbers of doctors and nurses who agreed with the use of the AN declined when life expectancy decreased [30]. These decisions can create ethical dilemmas and are related to feelings, thoughts and beliefs [31].

Sometimes, it is the patients themselves who do not wish to receive EN by NG tube [28]. Therefore, it is necessary to make an individualized decision, even though no other contraindications may be found. This respects the principles of autonomy, beneficence and non-maleficence [32]. Furthermore, the team of health care professionals looking after such patients should establish what the aims and benefits of such treatment are, whether these are achievable, and any possible damage that may be encountered [33]. In addition, the principle of autonomy recognizes the right and the capacity of a person to make their own personal decisions. Self-determination includes the right to reject EN, although this refusal may be difficult to understand for family members and healthcare professionals [3]. Perhaps the means to avoid ethical conflicts and future dilemmas is the use of anticipated instructions, where patients can reflect their decisions regarding future treatments or techniques, although the prevalence of patients who make use of such mechanisms is very low [34].

Among the limitations in this review are the lack of studies with a large enough sample to be able to describe the results, and the subjectivity of the results.

Although it is a difficult field of research, conducting higher-quality research could result in the provision of recommendations or guidance to aid patients and healthcare professionals in decision making.

The results obtained lead us to consider the need to create a clinical practice guide on the nutritional management of these patients, which includes the use of EN by NGT. Progress must be continued in education so that these differences do not exist, and such clinical practice is common to all nurses. The benefits and risks of the use of EN by NGT in these patients should be investigated, in order to provide evidence-based care. Clear evidence would help to reduce variability in the management of these patients.

## 5. Conclusions

There are not enough quality studies to provide evidence regarding the benefits for wellbeing and quality of life in patients under palliative care receiving EN through an NG tube.

For this reason, decision making in this field must be carried out individually, weighing the benefits and damages that they can cause in the quality of life of the patients.

## Figures and Tables

**Figure 1 nutrients-13-01562-f001:**
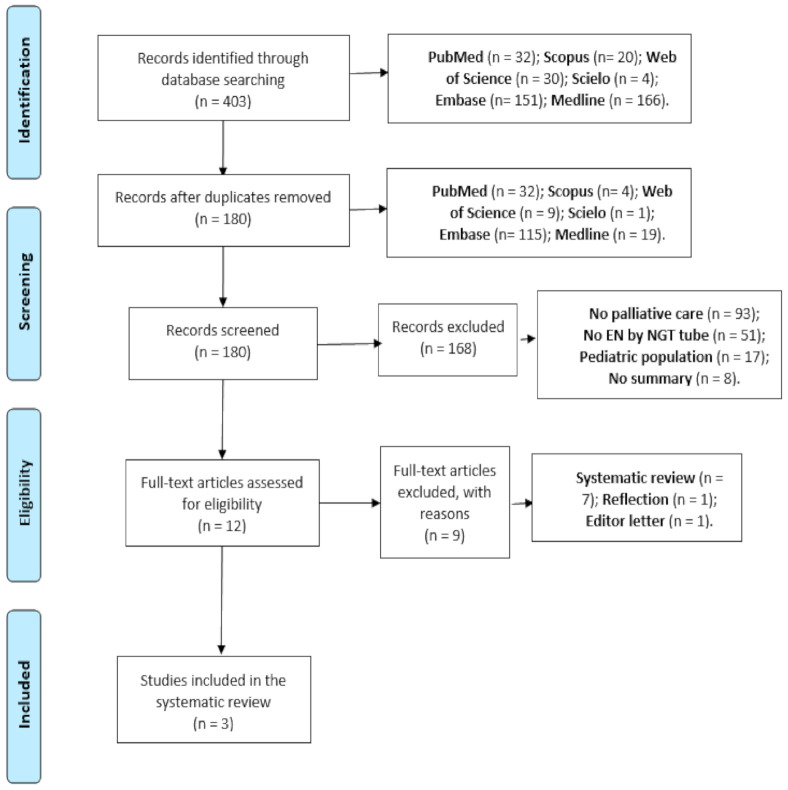
PRISMA (Preferred Reporting Items for Systematic Review and Meta-Analyses) diagram.

**Table 1 nutrients-13-01562-t001:** PICOS criteria (Population; Intervention; Comparison; Outcome; Study design).

P	Patients under Palliative Care
I	Enteral nutrition by nasogastric tube
C	The comparison could be made with any other type of dietary intervention or nutritional support.
O	Improving wellbeing and quality of life
S	Systematic review

**Table 2 nutrients-13-01562-t002:** Studies included in the systematic review.

Author, Year	Aim	Methodology	Results
Bentur N et al., 2015 [20].	To examine the prevalence of feeding tube use among older people with advanced dementia (OPAD) living in the community; to evaluate the characteristics, quality of care, and the burden on caregivers.	A cross-sectional survey of 117 caregivers of OPAD living in the community. They recorded clinical and care variables, treatment of symptoms at the end of life in dementia (SM-EOLD), evaluation of comfort in dementia at the end of life (CAD-EOLD) and satisfaction with the scales of care at the end of life with dementia (SWC-EOLD).	13% of patients carried NG tubes. The use of this type of device caused less diarrhea episodes than those subjects that did not use any feeding tube (6.6% vs. 32.5%) and more restrictions (60.0% vs. 9.9%, *p* < 0.05). Subjects with feeding tubes (NG tube or PEG) attended the emergency department at least once a day (40% vs. 34.2%, *p* < 0.05), and on more occasions (2.92 ± 1.68 vs. 1.6 ± 0.9 during the day and 2.9 ± 1.6 times compared to 1.4 ± 0.5 times during night time, *p* < 0.05). No statistical differences of significance were found between the use or non-use of feeding tubes in the scales of SM-EOLD, SWC-EOLD y CAD-EOLED, finding a difference in the wellbeing subscale of the last one, in subjects with or without feeding tubes, either NGT o PEG (6.9 ± 2.3 vs. 5.2 ± 2.0, *p* < 0.05, respectively), without finding differences between the types of feeding tubes.
Yang CW et al., 2015 [21].	Comparing clinical results of EN by tube and the placement of an esophageal stent in patients with malignant esophageal obstruction and a short life expectancy.	Retrospective observational study in 31 patients diagnosed with advanced-stage esophageal cancer, divided into three groups: patients with NG tube (*n* = 12), with esophageal stent, (*n* = 10) and patients with nutritional support but without oral intake (*n* = 9).	The average duration of hospital admissions was 19 days in the group of NGT, 12 days in the group of esophageal stents, and 39 days in the group without nutritional support (*p* = 0.01). The mean average survival after the diagnosis of malignant esophageal obstruction was 122 days in the group with NGT tube, 133 days in the group with esophageal stents and 51 days in those not receiving any nutritional support. The most common complication for the group using feeding tubes was pneumonia caused by aspiration (58%), although this was lower than in the group not receiving any nutritional support (100%).
Shinozaki et al., 2017 [22]	To examine the quality of life and functional state in terminal patients with brain and neck cancer.	Prospective and multicenter observational study with 11 oncology centers and hospitals in Japan. The survey EORTC QLQ-C15-PAL was used weekly formed by 15 items related with health wellbeing and quality of life. The sample was formed by 100 patients.	74.6% of patients required EN. Those with NGT showed longer hospital admissions than patients using PEG (64 compared to 21 days, *p* < 0.05). Patients using PEG presented shorter periods between the study prescription and death, compared to those fed by NGT. No significant difference was found in quality of life, between the starting point and week 3 of the study, among the different study groups.

## Data Availability

Data was collected from available literature. Data for doing the systematic review is available in the manuscript’s tables.

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
