# Peer review of "Enteral Nutrition by Nasogastric Tube in Adult Patients under Palliative Care: A Systematic Review"

_nutrients, 2021, doi:10.3390/nu13051562_

Round 1

Reviewer 1 Report

This study explores the ethical use of EN via feeding tube for patients undergoing palliative care. Their search strategy was inclusive and scientifically sound. I have a couple major points that I think would improve the manuscript and a few minor points.

First, most clinicians think of a nasogastric suction tube (large bore, stiff) when referring to NG tubes. I assume these studies involved nasogastric feeding tubes (12 or 14 Fr, soft). Did the reviewed studies specifically referred to them as feeding tubes?

That leads to the second query. What was the duration of feeding in these studies? A nasogastric feeding tube should not be used for much longer than 6 to 8 weeks due mechanical complications such as sinusitis, nasal septal erosions, etc. A nasogastric suction tube probably should not be used much longer than a week or so. Anticipated duration of feeding would also explain why some patients in these studies received their EN by PEG.

Minor comments

Abstract, line 24  delete “phase.”

Intro, line 36. Incomplete sentence. The following paragraph could be moved into the first paragraph.

Pg 4 Holland is misspelled, multcentric should be replaced with multicenter

P5, line 127. I assume that the 78.6% of patients that did not wish to receive EN by tube were not given EN by tube (I hope that is the case). Some clarity regarding this issue in the discussion would be useful.

Author Response

Dear reviewer,

Firstly, we appreciate the time dedicated to our manuscript, as well as the explanations that you ask for that help us to know doubts that a future reader may have, if the manuscript gets published.

Secondly, we answer to the questions that you have made, with aim of resolving doubts raised by our manuscript.

First, most clinicians think of a nasogastric suction tube (large bore, stiff) when referring to NG tubes. I assume these studies involved nasogastric feeding tubes (12 or 14 Fr, soft). Did the reviewed studies specifically referred to them as feeding tubes?

The reviewed studies refer to NGT, since when speaking of EN they are the ones used. Suction pumps do not fulfill this function and therefore do not appear in the reviewed studies.

That leads to the second query. What was the duration of feeding in these studies? A nasogastric feeding tube should not be used for much longer than 6 to 8 weeks due mechanical complications such as sinusitis, nasal septal erosions, etc. A nasogastric suction tube probably should not be used much longer than a week or so. Anticipated duration of feeding would also explain why some patients in these studies received their EN by PEG.

Eso lleva a la segunda consulta. ¿Cuál fue la duración de la alimentación en estos estudios? Una sonda de alimentación nasogástrica no debe usarse durante mucho más de 6 a 8 semanas debido a complicaciones mecánicas como sinusitis, erosiones del tabique nasal, etc. Una sonda de succión nasogástrica probablemente no debe usarse por más de una semana aproximadamente. La duración anticipada de la alimentación también explicaría por qué algunos pacientes en estos estudios recibieron su NE por PEG.

Duration is not present in any of the reviewed studies. Sometimes the use of these devices is carried out in the last few days of life, and this makes us think that the duration of the NGT is not very long in time. Although, as can be seen in the review, there are not enough studies in this field and they are of low quality.

Minor comments

Abstract, line 24 delete “phase.”

The abstract has been modified, because a large part of the manuscript has been modified at the request of the different reviewers.

Intro, line 36. Incomplete sentence. The following paragraph could be moved into the first paragraph.

The introduction has been modified with the aim to improve the contextualization and the goal of the topic reviewed, covering in a more precise manner, the need to carry out this review. Furthermore, after taking into account the comments made by the reviewers; we have again reviewed the table of selected articles, and the studies by Hendricks et al and Amano et al. have been excluded, because they solely studied the decision-making and not whether this technique would improve health wellbeing and the quality of life. The discussion has also been modified to improve the details of the review.

Pg 4 Holland is misspelled, multcentric should be replaced with multicenter

Modification made.

P5, line 127. I assume that the 78.6% of patients that did not wish to receive EN by tube were not given EN by tube (I hope that is the case). Some clarity regarding this issue in the discussion would be useful.

Paragraph modified to clarify the statement.

Once again, we appreciate the time and attention dedicated to our manuscript. We really hope we have reached your expectations, with the modifications made and that the explanations to those that we have not modified be considered as appropriate.

Kind regards.

Reviewer 2 Report

This review aimed to know the current recommendations regarding the management of EN by NGT in patients receiving palliative care based on the articles published between 2015 and 2020. The results shows no clear recommendation based on the currently available evidence.

I agree that the use of EN for the patients in palliative care is subject to be discussed from the viewpoint in both ethical aspect and merit for improving quality of life. However, there are some concerns need to be addressed.

#1 The aim of this review may be unclear. The Cochrane systematic review cited by the authors dealt with any types of artificial nutrition, but this study only focused on the EN by NG tube. Why the authors limited to NG tube and did not include other artificial nutrition?

#2 The methods was not well systematically performed. For example; outcome measures were not specified despite “improving quality of life” was clearly stated in PICOS form; although the method of nutrition support was limited to NG tube, the authors did not use the term “naso-gastric tube” or like that in literature search; the assessment method of the literature was not described. Who and how assess the quality of each literatures?

#3 Some articles such as Yang’s study did not contain the outcome of interest (quality of life). Additionally, several articles included not only the patients with NG tube but also those with PEG. I wonder if the assessor(s) correctly chose the relevant articles. I guess the results did not answer the research question due to insufficient methodology of systematic reviews.

#4 Table 3 needs to be improved. It is preferred to eliminate the Titles and Aims, rather added some information such as the exposure and outcome of interest and results with precision (e.g., 95%CI or p-values).

Author Response

Dear reviewer,

Firstly, we appreciate the time dedicated to our manuscript, as well as the explanations that you ask for that help us to know doubts that a future reader may have, if the manuscript gets published.

Secondly, we answer to the questions that you have made, with aim of resolving doubts raised by our manuscript.

This review aimed to know the current recommendations regarding the management of EN by NGT in patients receiving palliative care based on the articles published between 2015 and 2020. The results show no clear recommendation based on the currently available evidence.

There is no clear evidence to help us write a specific recommendation, as similarly has occurred in previous reviews on this topic. This makes us to consider that further research on this field should continue.

I agree that the use of EN for the patients in palliative care is subject to be discussed from the viewpoint in both ethical aspect and merit for improving quality of life. However, there are some concerns need to be addressed.

#1 The aim of this review may be unclear. The Cochrane systematic review cited by the authors dealt with any types of artificial nutrition, but this study only focused on the EN by NG tube. Why the authors limited to NG tube and did not include other artificial nutrition?

We focused on the nasogastric tube because during our clinical practice it was the nutritional access route that created the most doubts among professionals and family members. In addition, patients in palliative care reported that the placement of this nasogastric tube affected their daily life, especially because it looked be aesthetically and it was painful and distressing.

Therefore, the goal of this review was to give answers to the needs of our patients.

#2 The methods were not well systematically performed. For example; outcome measures were not specified despite “improving quality of life” was clearly stated in PICOS form; although the method of nutrition support was limited to NG tube, the authors did not use the term “nasogastric tube” or like that in literature search; the assessment method of the literature was not described. Who and how assess the quality of each literatures?

The introduction was modified to adapt the topics of quality of life, encompassing the physical, psychological, and spiritual aspects. The improvement in these three spheres could benefit the quality of life, meaning, hospital stays, overload of caregivers, etc...

Due to the possible misinterpretation of the formula PICOS, the authors have decided to modified it for the next statement “Improvement in the physical, psychosocial, spiritual and quality of life aspects”

Before commencing the search, the MeSH terms were considered and “nasogastric tube” did not appear in the list, however it was considered as “Entry Terms” which was subordinated to the MeSH “Enteral Nutrition”. Although it involved more work and time reviewing articles by the researchers, we thought that we would not miss any information by using the Entry Terms.

As it appears in the methodology, two screenings were carried out. For the second, the STROBE statement (Strengthening the Reporting of Observational studies in Epidemiology) was used for observational studies and the CONSORT guide (Consolidated Standards of Reporting Trials) for randomized clinical trials.

#3 Some articles such as Yang’s study did not contain the outcome of interest (quality of life). Additionally, several articles included not only the patients with NG tube but also those with PEG. I wonder if the assessor(s) correctly chose the relevant articles. I guess the results did not answer the research question due to insufficient methodology of systematic reviews.

The table has been modified in order to represent more accurately the data obtained.  The aim has been modified to "improvement of quality of life and health status" since, although quality of life is not measured directly, the patient's health status influences the perception of quality of life, thus it is presented in the introduction.

Furthermore, after taking into account the comments made by the reviewers; we have again reviewed the table of selected articles, and the studies by Hendricks et al and Amano et al. have been excluded, because they solely studied the decision-making and not whether this technique would improve health wellbeing and the quality of life. The discussion has also been modified to improve the details of the review.

#4 Table 3 needs to be improved. It is preferred to eliminate the Titles and Aims, rather added some information such as the exposure and outcome of interest and results with precision (e.g., 95%CI or p-values).

Following your comments, the table has been modified, presenting more data of interest to the reader.

In addition, the spelling has been reviewed again with qualified personnel in order to solve the spelling problems.

Once again, we appreciate the time and attention dedicated to our manuscript. We really hope we have reached your expectations, with the modifications made and that the explanations to those that we have not modified be considered as appropriate.

Kind regards.

Reviewer 3 Report

An interesting review study but the introduction was not able to really bring the problem in a general context and was not able to justify the realization of this review. The introduction is in general not convincing.

In general,many details are missing.

Author Response

Dear reviewer,

Firstly, we appreciate the time dedicated to our manuscript, as well as the explanations that you ask for that help us to know doubts that a future reader may have, if the manuscript gets published.

Secondly, we answer to the questions that you have made, with aim of resolving doubts raised by our manuscript.

An interesting review study but the introduction was not able to really bring the problem in a general context and was not able to justify the realization of this review. The introduction is in general not convincing.

The introduction has been modified with the aim to improve the contextualization and the goal of the topic reviewed, covering in a more precise manner, the need to carry out this review. Furthermore, after taking into account the comments made by the reviewers; we have again reviewed the table of selected articles, and the studies by Hendricks et al and Amano et al. have been excluded, because they solely studied the decision-making and not whether this technique would improve health wellbeing and the quality of life.

In generalmany details are missing.

Once again, we appreciate the time and attention dedicated to our manuscript. We really hope we have reached your expectations, with the modifications made and that the explanations to those that we have not modified be considered as appropriate.

Kind regards.

Round 2

Reviewer 2 Report

The authors addressed the concerns raised in the last round of review and several flaws are appropriately corrected. However, there are still some concerns.

#1 The authors replied ‘Due to the possible misinterpretation of the formula PICOS, the authors have decided to modified it for the next statement “Improvement in the physical, psychosocial, spiritual and quality of life aspects’. It is unclear what is meant by “physical, psychosocial, spiritual and quality of life aspects" and how the authors confirm that these components are appropriately measured, analyzed, and interpretable in each of the selected references. Please clarify.

#2 Outcome measures on Table 1 is inconsistent to the authors’ reply #1 (“Improving wellbeing and quality of life”). Please address.

#3 Although PICOS form was used prior to searching literature in this study, the terms regarding outcomes such as “quality of life” or “physical function” were not used for search strategy. Please consider add the words regarding outcome measures to searching, if possible.

#4 In Figure 1, text in the right bottom rectangle (“Pubmed (n=2); Embase (n=1)”) would make no sense. Please consider deleting it.

Author Response

Dear reviewer,

Thank you again for reviewing our manuscript with the modifications made after the first revision. We have answered the new clarifications requested, hoping that we have done so satisfactorily.

The clarifications requested and the authors' response are detailed below.

The authors addressed the concerns raised in the last round of review and several flaws are appropriately corrected. However, there are still some concerns.

#1 The authors replied ‘Due to the possible misinterpretation of the formula PICOS, the authors have decided to modified it for the next statement “Improvement in the physical, psychosocial, spiritual and quality of life aspects’. It is unclear what is meant by “physical, psychosocial, spiritual and quality of life aspects" and how the authors confirm that these components are appropriately measured, analyzed, and interpretable in each of the selected references. Please clarify.

#2 Outcome measures on Table 1 is inconsistent to the authors’ reply #1 (“Improving wellbeing and quality of life”). Please address.

It is a misprint in the answers to the first clarifications requested, due to a problem in the initial drafting and the subsequent translation. In the PICOS formula, "improving well-being" was added, where we refer to the
improvement of physical, psychosocial and spiritual conditions that provide comfort, as well-being is defined.

This definition encompasses terms such as: decrease in hospital stays, visit the emergency department less, decrease in restrictions and possible complications improve the well-being of these patients. These concepts appear in the results.

#3 Although PICOS form was used prior to searching literature in this study, the terms regarding outcomes such as “quality of life” or “physical function” were not used for search strategy. Please consider add the words regarding outcome measures to searching, if possible.

When the PICOS formula was modified, a search was carried out with the new terms “Quality of life” and “Improving wellbeing” and “wellbeing”, using different search equations, not increasing the results obtained in the
search.

#4 In Figure 1, text in the right bottom rectangle (“Pubmed (n=2); Embase (n=1)”) would make no sense. Please consider deleting it.

This part has been deleted as suggested.

Kind regards